# Impact of Sacubitril/Valsartan on Cardiac Reverse Remodeling in Patients with Heart Failure Undergoing Cardiac Resynchronization Therapy

**DOI:** 10.3390/diseases14010006

**Published:** 2025-12-27

**Authors:** Tariel Atabekov, Irina Silivanova, Irina Kisteneva, Sergey Krivolapov, Roman Batalov, Sergey Popov

**Affiliations:** Cardiology Research Institute, Tomsk National Research Medical Center, Russian Academy of Sciences, Kievskaya St., 111a, 634012 Tomsk, Russia

**Keywords:** heart failure, cardiac resynchronization therapy, sacubitril/valsartan, cardiac reverse remodeling

## Abstract

**Background/Objectives:** Many heart failure (HF) patients exhibit a suboptimal response to cardiac resynchronization therapy (CRT). This study investigated whether sacubitril/valsartan, a drug known to beneficially impact cardiac remodeling, could improve outcomes for patients undergoing CRT implantation. **Methods:** In this single-center, observational study, 90 HF patients (left ventricular ejection fraction [LVEF] ≤ 35%) receiving a CRT-defibrillator were stratified into a sacubitril/valsartan group (*n* = 39) and a control group (*n* = 51). The primary endpoint was a CRT response at 12 months, defined as improvement in New York Heart Association (NYHA) class, left ventricular reverse remodeling (≥15% reduction in left ventricular end-systolic volume [LVESV] or ≥5% improvement in LVEF), and freedom from HF hospitalization. **Results:** The sacubitril/valsartan group had a significantly higher CRT response rate (87.2% vs. 64.7%, *p* = 0.016). They also showed greater improvement in the 6 min walk test (*p* = 0.013), NYHA class (*p* = 0.017), reduction in LVESV (*p* = 0.025), and QRS duration (*p* = 0.005). Multivariable analysis confirmed sacubitril/valsartan as an independent predictor of CRT response (OR = 4.43; 95% CI: 1.33–14.71; *p* = 0.015). **Conclusions:** In this study of HF patients receiving CRT, sacubitril/valsartan was independently associated with superior reverse remodeling, enhanced electrical resynchronization, and a higher rate of CRT response. These findings suggest a potential synergistic role for sacubitril/valsartan in optimizing post-CRT outcomes; however, as this was an observational study, they should be considered hypothesis-generating and require validation in larger, randomized controlled trials.

## 1. Introduction

Cardiac resynchronization therapy (CRT) represents a cornerstone in the management of patients with heart failure (HF) and electrical dyssynchrony, conferring significant benefits in morbidity and mortality by restoring synchronous ventricular contraction [1]. However, a substantial proportion of patients exhibit a suboptimal response, failing to achieve the desired degree of cardiac reverse remodeling [2]. This process of reverse remodeling, the partial normalization of cardiac structure and function, is intrinsically linked to the complex molecular mechanisms in cardiac remodeling. These mechanisms involve a cascade of maladaptive signaling pathways, including neurohormonal activation, fibrotic changes, and alterations in calcium handling, which drive the progression of the underlying cardiomyopathy [3].

A critical and perilous consequence of this adverse remodeling is the heightened susceptibility to life-threatening cardiac arrhythmias. The arrhythmogenic substrate in the failing heart is multifaceted, stemming from myocardial fibrosis creating regions of slow conduction and a key role played by the dysregulation of ventricular repolarization [3,4]. This dysregulation is frequently mediated by pathological alterations in ion channel expression and function, particularly the downregulation of the transient outward potassium current and the inward rectifier potassium current, which promote action potential prolongation and early afterdepolarizations [5].

The advent of angiotensin receptor-neprilysin inhibitors (ARNIs) has marked a paradigm shift in HF therapy. Sacubitril/valsartan, by concurrently inhibiting the angiotensin II receptor and neprilysin, augments beneficial vasoactive peptides such as natriuretic peptides while suppressing the deleterious renin–angiotensin–aldosterone system [6,7]. While the superiority of ARNIs over enalapril in reducing cardiovascular death and HF hospitalizations is well-established [8], its specific electrophysiological and structural impact on patients receiving CRT device-based therapy remains less defined. However, emerging evidence shows that ARNIs therapy holds promise in enhancing CRT response, particularly in HF patients with reduced left ventricular ejection fraction (LVEF) who are non-responders [9].

Given the interplay between neurohormonal modulation, reverse remodeling, and arrhythmogenesis, we hypothesize that sacubitril/valsartan would potentiate the benefits of CRT. This study therefore aims to investigate the impact of sacubitril/valsartan on the extent of cardiac reverse remodeling in patients with HF undergoing CRT, with a specific focus on its potential to modify the arrhythmic substrate through the modulation of pathways central to cardiomyopathy progression and ion channel homeostasis.

## 2. Materials and Methods

### 2.1. Study Design and Participants

This single-center, non-randomized, prospective, observational study enrolled consecutive patients with LVEF ≤ 35% who underwent CRT-defibrillator (CRT-D) implantation between February 2021 and July 2025.

Key exclusion criteria were as follows: age < 18 years; New York Heart Association (NYHA) class I or IV HF; hypertrophic cardiomyopathy; persistent and permanent atrial fibrillation (AF); prior or unsuccessful CRT-D implantation; incomplete clinical, echocardiographic, or medication data at 12-month follow-up; and sacubitril/valsartan treatment initiated >3 months before implantation.

The final study population comprised patients with NYHA class II or III HF, sinus rhythm, LVEF ≤ 35%, and a QRS duration ≥ 130 ms (with left bundle branch block [LBBB]) or ≥150 ms (with non-LBBB morphology), despite optimal medical therapy. Patients were stratified into two cohorts based on sacubitril/valsartan treatment: those receiving the drug (1st group) and those not receiving it (2nd group). In the sacubitril/valsartan group, the median dose administered both at baseline (before CRT-D implantation) and at the 12-month follow-up was 100.0 mg, with an interquartile range of [100.0; 200.0]. This indicates that the cohort was predominantly maintained on a stable regimen of 100 mg or 200 mg twice daily throughout the study period. The protocol allowed for individualized titration based on patient tolerance, but the median dose remaining constant suggests that for most patients, the dose established prior to implantation was well-tolerated and maintained long-term.

### 2.2. Consent

All patients provided written informed consent for both participation and publication of their data. The study was conducted in accordance with the Declaration of Helsinki and Good Clinical Practice guidelines and was approved by the Local Ethics Committee of the Cardiology Research Institute (protocol No 219 [26 October 2021] and 240 [15 February 2023]). Most participants were enrolled from registered clinical trials (NCT03667989 and NCT05769036).

### 2.3. Data Collection and Methods

At baseline, all patients underwent a comprehensive assessment, which included the collection of demographic data, medical history, cardiovascular risk factors, and pharmacological treatments. Patients also completed the Minnesota Living with Heart Failure Questionnaire (MLWHFQ) and the European Quality of Life Questionnaire (EuroQol EQ-5D). A full physical examination was performed, incorporating a 6 min walk test (6MWT), electrocardiography (ECG), Holter monitoring, transthoracic and speckle-tracking echocardiography (TTE and STE), coronary angiography, and blood analysis. This assessment protocol was repeated at the 12-month follow-up after CRT-D implantation, at which time device interrogation data were also collected. All patients received standard guideline-directed medical therapy for HF throughout the study [10].

Ischemic etiology of HF was defined by the presence of significant coronary artery disease (≥50% stenosis in a major coronary artery on angiography or computed tomography-angiography), a history of myocardial infarction, or prior revascularization.

#### 2.3.1. Six-Minute Walk Test

Functional capacity was assessed according to NYHA class, which was determined based on the 6 min walking distance as follows:->551 m: NYHA class I (no apparent symptoms);-426–550 m: NYHA class II (mild symptoms);-301–425 m: NYHA class III (marked symptoms);-151–300 m: NYHA class IV (severe symptoms);-≤150 m: NYHA class IV.

#### 2.3.2. Electrocardiogram Analysis

Standard 12-lead ECGs were recorded at a speed of 25 mm/s and an amplitude of 10 mm/mV before and after device implantation. A blinded investigator analyzed the ECGs for the following parameters: bundle branch block morphology; Strauss criteria compliance for LBBB [11]; PQ interval; alpha angle; and QRS duration at baseline and 12-month follow-up. The corrected QT interval was adjusted for bundle branch block or biventricular pacing (BP) [12,13]. Additional qualitative assessments included the presence of QRS notching or slurring in leads I, aVL, V5, and V6, and QS or rS patterns in V1 and V2.

#### 2.3.3. Echocardiographic Acquisition and Analysis

Transthoracic echocardiography was performed at baseline and 12 months post-implantation using a Philips HD15 PureWave system (Philips Ultrasound, Inc., 22100 Bothell Everett Highway, Bothell, WA 98021-8431, USA). All analyses were conducted by investigators blinded to patient response status, in accordance with current guidelines [14]. The comprehensive protocol included chamber quantification (assessment of volume-dimensional parameters for all cardiac chambers), function assessment (measurement of LVEF and evaluation of mitral, tricuspid, and aortic valve function and ventricular contractility).

#### 2.3.4. Speckle-Tracking Echocardiography

Speckle-tracking echocardiography analysis for intracardiac hemodynamics was performed using an Affinity 70 CV ultrasound system (Philips Ultrasound, Inc., 22100 Bothell Everett Highway, Bothell, WA 98021-8431, USA) with high-frequency phased-array transducers (3–8 MHz). Data were acquired in standard B-mode, M-mode, and Doppler modes at a frame rate of 60–110 fps during brief breath-holds, where possible. Native data were stored for offline analysis in the Q-lab 15.7 software package. The global longitudinal strain parameter was derived.

#### 2.3.5. CRT-D Implantation and Programming

The active-fixation defibrillation lead was implanted in either the septum or apex of the right ventricle, and the active-fixation atrial lead was positioned in the right atrial appendage or the high right atrium. A passive-fixation quadripolar left ventricular (LV) lead was delivered via the coronary sinus to a lateral branch (posterolateral, lateral, or anterolateral). The specific lead models and final positions were selected at the implanting physician’s discretion, following a standard transvenous approach under fluoroscopic guidance. Lead placement was confirmed using fluoroscopic imaging (postero-anterior and left anterior oblique views) and intra-operative threshold testing. A pacing system analyzer was used to measure lead capture thresholds, impedance and sensing amplitudes.

For device programming, a standardized protocol based on international guidelines was employed [15]. A monitoring zone was set at 140–170 bpm (requiring >50 cycles) without therapy. The ventricular tachyarrhythmia (VT) zone was programmed for 170–200 bpm (30 cycles) and was configured to deliver antitachycardia pacing (ATP) (including at least one burst and one ramp sequence) followed by a submaximal shock. The ventricular fibrillation zone was set for 201–240 bpm (12 cycles), with ATP delivered during charging and a maximum-energy shock as the first therapy.

### 2.4. Definition of CRT Response Criteria

Patients were classified as CRT responders if they demonstrated a comprehensive response at the 12-month follow-up. This required an improvement in NYHA class of at least one grade, evidence of left ventricular reverse remodeling (defined as a ≥15% reduction in end-systolic volume or a ≥5% improvement in ejection fraction), and survival without any heart failure-related hospitalizations [16,17]. Patients who met the pre-specified CRT response criteria were classified as “responders,” while those who did not were classified as “non-responders.”

### 2.5. Study Endpoints

The primary endpoint was the response to CRT at 12 months. Secondary endpoints included cardiovascular mortality, HF-related hospitalization, arrhythmic events recorded by the CRT device, and appropriate implantable cardioverter-defibrillator (ICD) therapy.

### 2.6. Statistical Analysis

Categorical data are presented as numbers and percentages. Continuous variables are summarized as the mean ± standard deviation or median with interquartile range [Q1; Q3], based on their distribution. Normality was assessed using the Kolmogorov–Smirnov and Shapiro–Wilk tests. Group comparisons for continuous variables were performed using the Student’s *t*-test (for normally distributed data) or the Mann–Whitney U test (for non-normally distributed data). The Wilcoxon test was used for dependent samples. Categorical variables were compared using the Chi-square or Fisher’s exact test, as appropriate.

To identify predictors of CRT response (primary endpoint), we conducted a stepwise logistic regression analysis. First, a univariable analysis was performed to test all clinical variables, including sacubitril/valsartan treatment, for association with the primary endpoint. Variables with a significance of *p* < 0.05 in the univariable analysis were included as candidates in a multivariable model. Multicollinearity among these variables was assessed and addressed prior to model fitting. The model’s goodness-of-fit was evaluated using the Hosmer–Lemeshow test. Results from our regression analysis are presented as odds ratio (OR) with 95% confidence intervals (CI). The area under the curve (AUC) was calculated to evaluate the discriminatory ability of the variables associated with CRT response.

Survival analysis was performed using the Kaplan–Meier method, and outcomes-free survival was compared between groups stratified by sacubitril/valsartan treatment using the log-rank test.

All analyses were conducted using Statistica (version 14.1.0., StatSoft Inc., Tulsa, OK, USA) and Medcalc (version 19.2.6, MedCalc Software, Ostend, Belgium), with a two-sided *p*-value < 0.05 considered statistically significant.

## 3. Results

### 3.1. Study Population Characteristics

The patient selection process is detailed in Figure 1. Of 119 screened patients, 90 were enrolled in the study after the exclusion of 29 individuals (25 with incomplete follow-up and 4 with persistent AF). The cohort was divided into a sacubitril/valsartan group (*n* = 39, 43.3%) and a control group not receiving this treatment (*n* = 51, 56.7%).

As shown in Table 1, the groups were well-matched at baseline. There were no significant differences in demographic profiles, clinical history, symptomatology (assessed by NYHA class), or baseline cardiac structure and function (via ECG, TTE and STE). Post-procedural characteristics, including LV lead position and 12-month BP rate, were also similar (Table 1 and Table 2).

Regarding cardiac medication at admission, the groups were comparable except for the use of angiotensin-converting enzyme inhibitors (ACEIs) and angiotensin II receptor blockers (ARBs). As expected, a formal statistical difference was observed (*p* < 0.001 for both), as patients in the sacubitril/valsartan group did not concurrently take ACEIs or ARBs. A minor but significant difference was also noted in baseline QRS duration (179.8 ± 24.7 ms vs. 168.0 ± 19.5 ms, *p* = 0.019).

We systematically evaluated the safety profile of sacubitril/valsartan over the 12-month study period. Critically, no significant adverse events, specifically, no episodes of symptomatic hypotension or clinically relevant deterioration of kidney function, were reported that led to drug discontinuation. Objective hemodynamic and laboratory data support this excellent safety profile. Systolic blood pressure remained stable (118.5 ± 8.5 mmHg before CRT-D vs. 119.1 ± 5.8 mmHg at 12 months; *p* = 0.010). Diastolic blood pressure also showed no clinically significant change (73.8 ± 4.9 mmHg before vs. 74.5 ± 3.8 mmHg at 12 months; *p* = 0.384). Serum creatinine levels showed a slight but statistically significant decrease over the study period (91.9 ± 22.6 µmol/L before CRT-D vs. 89.6 ± 24.0 µmol/L at 12 months; *p* < 0.001), indicating no adverse impact on renal function.

### 3.2. Clinical and Echocardiographic Outcomes at 12-Month Follow-Up After CRT

At baseline, the two groups were comparable in terms of EuroQol EQ-5D, MLWHFQ, and 6MWT scores (Table 2). After 12 months of CRT, both groups demonstrated significant improvement in all three measures. In the sacubitril/valsartan group, the EuroQoL EQ-5D score improved from 57.3 ± 10.3 to 76.6 ± 9.6 (*p* < 0.001), and the MLWHFQ score decreased from 59.6 ± 17.9 to 24.7 ± 14.8 (*p* < 0.001). Similarly, the control group (no sacubitril/valsartan) showed improvement in EuroQoL EQ-5D (from 58.5 ± 9.5 to 75.1 ± 9.2; *p* < 0.001) and MLWHFQ (from 59.4 ± 17.7 to 28.3 ± 18.8; *p* < 0.001). The improvement in the 6MWT was more pronounced in the sacubitril/valsartan group, with 97.4% (38/39) of patients showing improvement compared to 78.4% (40/51) in the control group (*p* = 0.009). The absolute improvement in 6MWT distance was also significantly greater in the sacubitril/valsartan group (*p* = 0.013) (Figure 2A).

A significant shift in NYHA functional class was observed in both groups at 12 months, with an increase in class I and a decrease in class III patients (both *p* < 0.001). However, the sacubitril/valsartan group had a significantly higher proportion of patients in NYHA class I (*p* = 0.017) and a lower proportion in class III (*p* = 0.041) compared to the control group (Figure 3).

Echocardiographic parameters improved significantly in both groups, with LVEF increasing and left ventricular end-systolic volume (LVESV) decreasing (both *p* < 0.001) (Figure 2C,D). While the change in LVEF (∆LVEF) was not significantly different between groups (*p* = 0.079), the reduction in LVESV (∆LVESV) was significantly greater in the sacubitril/valsartan group (*p* = 0.025) (Figure 4A).

Based on the study’s CRT response criteria, 87.2% (34/39) of patients in the sacubitril/valsartan group were classified as responders, compared to 64.7% (33/51) in the control group (*p* = 0.016) (Figure 5).

Finally, QRS duration decreased significantly in both groups after CRT-D implantation (*p* < 0.001) (Figure 2B), with a significantly greater reduction (∆QRS) observed in the sacubitril/valsartan group (*p* = 0.005) (Figure 4B).

### 3.3. Risk Stratification Analysis

Receiver operating characteristic analysis identified sacubitril/valsartan treatment, baseline creatinine level, and baseline left ventricular end-systolic dimension (LVESD) as the parameters with the strongest discriminative power for predicting CRT response, with AUC values of 0.645 (95% CI: 0.537–0.743), 0.661 (95% CI: 0.554–0.758), and 0.675 (95% CI: 0.568–0.770), respectively (Figure 6).

In univariable logistic regression, several factors were significantly associated with CRT response (Figure 7). These included sacubitril/valsartan treatment (OR = 3.71; 95% CI: 1.23–11.14; *p* = 0.019), baseline creatinine (OR = 0.97; 95% CI: 0.95–0.99; *p* = 0.027), baseline LVESD (OR = 0.91; 95% CI: 0.85–0.98; *p* = 0.015), ACEIs treatment (OR = 0.35; 95% CI: 0.13–0.92; *p* = 0.035), baseline QRS notching in lead V5 (OR = 2.96; 95% CI: 1.11–7.91; *p* = 0.029), and baseline asynchronous contraction of the interventricular septum (OR = 4.79; 95% CI: 1.02–22.34; *p* = 0.046).

A multivariable model was constructed, adjusting for non-ischemic HF etiology, age, female gender, baseline QRS duration, LV lead lateral position, percent of BP at 12 months, baseline LVEF and LVESV, Strauss criteria for LBBB, and paroxysmal AF. In this adjusted analysis, only three variables remained independent predictors of CRT response: sacubitril/valsartan treatment (OR = 4.43; 95% CI: 1.33–14.71; *p* = 0.015), baseline creatinine (OR = 0.97; 95% CI: 0.95–0.99; *p* = 0.037), and baseline LVESD (OR = 0.90; 95% CI: 0.83–0.97; *p* = 0.010).

### 3.4. Overall Survival

During the 12-month follow-up, there were no cardiovascular deaths related to HF. Over the 12-month follow-up period, HF hospitalization was a rare event, occurring in one patient (2.4%) in the sacubitril/valsartan group and one patient (1.9%) in the control group. A total of 11 patients (28.2%) in the sacubitril/valsartan group and 11 patients (21.5%) in the control group experienced CRT-D registered arrhythmic events (*p* = 0.470). Kaplan-Meier estimates for these events are shown in Figure 8. The majority of these events were nonsustained VT, which terminated spontaneously (10/11 in each group). One patient in each group (2.4% vs. 1.9%, *p* = 0.848) received appropriate ICD therapy for sustained VT.

## 4. Discussion

The principal finding of this study is that treatment with sacubitril/valsartan in patients with HF undergoing CRT-D implantation is independently associated with a significantly higher rate of positive response to resynchronization therapy. After 12 months of follow-up, patients receiving sacubitril/valsartan demonstrated superior improvements in functional capacity, greater left ventricular reverse remodeling, and a more pronounced electrical response, as evidenced by a greater reduction in QRS duration. Importantly, in multivariable analysis, sacubitril/valsartan therapy emerged as a powerful independent predictor of CRT response, alongside established factors such as baseline renal function and left ventricular size.

Our findings align with and extend the growing body of evidence supporting the synergistic effects of sacubitril/valsartan with device-based therapy. The observed odds ratio of 4.43 for CRT response underscores a potent effect. The greater reduction in LVESV (∆LVESV) in the sacubitril/valsartan group points to a more robust effect on the key structural molecular mechanisms in cardiac remodeling. While both groups showed improved LVEF, the significant advantage in LVESV reduction suggests that sacubitril/valsartan may facilitate a deeper regression of the maladaptive processes that define the progressive cardiomyopathy of HF. This is biologically plausible, as the dual neprilysin inhibition and angiotensin receptor blockade offers a more comprehensive neurohormonal modulation than conventional renin–angiotensin–aldosterone system inhibition alone [6]. By decreasing natriuretic peptides and concurrently suppressing angiotensin II, sacubitril/valsartan targets multiple pathways involved in hypertrophy, fibrosis, and adverse remodeling, thereby creating a more favorable substrate for CRT to exert its mechanical benefits [18,19,20].

A novel and intriguing observation from our study is the significantly greater reduction in QRS duration (∆QRS) in the sacubitril/valsartan group. CRT works by ameliorating electrical dyssynchrony, and a greater QRS narrowing post-implantation has been linked to superior outcomes [21]. The observed reduction in QRS duration in our study suggests a potential effect on myocardial conduction. While our data cannot establish a direct causal mechanism, we hypothesize that this effect may be mediated, at least in part, by the known anti-fibrotic properties of sacubitril/valsartan. This hypothesis is supported by clinical evidence. For instance, in the PROVE-HF study [22], treatment with sacubitril/valsartan in humans was associated with significant reductions in serum biomarkers of collagen turnover, which correlated with reverse cardiac remodeling. Therefore, it is a plausible, though not yet proven, hypothesis that a reduction in interstitial fibrosis could ameliorate conduction heterogeneity and contribute to the QRS narrowing we observed, representing an exciting area for future mechanistic investigation.

Furthermore, our study provides preliminary clinical insights into the complex relationship between reverse remodeling and arrhythmogenesis. The progression of cardiomyopathy is frequently complicated by an increased burden of cardiac arrhythmias, often stemming from pathological alterations in ion channel expression and function, such as the downregulation of potassium currents leading to action potential prolongation [5]. While our study was not powered to detect differences in hard arrhythmic endpoints, the numerical, though non-significant, reduction in overall arrhythmic events and appropriate ICD therapies in the sacubitril/valsartan group is hypothesis-generating. It is conceivable that the profound reverse remodeling induced by sacubitril/valsartan, by reducing chamber dilation and fibrosis, modifies the macro-reentrant substrate for ventricular tachycardia [3,4]. Additionally, experimental models suggest that sacubitril/valsartan may directly or indirectly stabilize electrophysiology by mitigating calcium-handling abnormalities and normalizing the expression of key ion channel proteins, thereby reducing triggers for arrhythmias like early afterdepolarizations [23]. The similar rate of HF hospitalizations and absence of cardiovascular mortality in both groups over 12 months are encouraging and may reflect the overall efficacy of modern HF care, including CRT-D, in a relatively stable cohort.

Beyond the significant impact of sacubitril/valsartan, our multivariable analysis reaffirmed the prognostic importance of two fundamental, baseline patient characteristics: renal function, as reflected by serum creatinine level, and the degree of left ventricular dilatation, measured by LVESD. The inverse association of both these variables with CRT response highlights the critical influence of the pre-existing cardiorenal and structural substrate on the likelihood of benefiting from device therapy.

The independent association between higher baseline creatinine levels and a lower probability of CRT response is consistent with a substantial body of literature. Impaired renal function is a well-established marker of adverse prognosis in HF and is integral to the cardiorenal syndrome [24]. The pathophysiological link likely involves several mechanisms. First, chronic kidney disease (CKD) is associated with a state of chronic inflammation, increased oxidative stress, and neurohormonal activation that can accelerate myocardial fibrosis and adverse remodeling, creating a substrate less amenable to reverse remodeling [25]. Second, patients with renal dysfunction often have a higher burden of comorbidities, including hypertension and diabetes, which can contribute to a more diffuse and irreversible myocardial disease. Third, CKD can exacerbate electrolyte imbalances and increase the risk of arrhythmias, potentially confounding the clinical course and response to therapy [25,26]. Our findings suggest that even modest elevations in serum creatinine, within a range typical for a HF population, identify a patient subgroup with a more advanced disease state, in whom the maladaptive pathways may be less reversible, even with aggressive interventions like CRT.

Similarly, the strong inverse correlation between baseline LVESD and CRT response underscores the concept of a “point of no return” in the structural remodeling process. A larger LVESD signifies more advanced ventricular dilation and a greater loss of viable, contractile myocardium, often replaced by fibrous tissue [27]. CRT primarily improves cardiac function by coordinating the contraction of viable myocardial segments. In a grossly dilated ventricle with extensive scarring, the potential for functional resynchronization is inherently limited. The mechanical inefficiency of a spherical, dilated ventricle, governed by the law of Laplace (where wall stress is directly proportional to ventricular radius and inversely proportional to wall thickness), is more difficult to reverse [28]. Our results align with prior studies that have identified left ventricular end-systolic volume as a powerful predictor of CRT outcome [29]. The use of LVESD in our study, a linear measurement readily available on standard echocardiography, provides a clinically practical and similarly potent marker of this unfavorable structural substrate.

The coexistence of these two factors, renal impairment and severe ventricular dilation, in a single patient likely identifies a phenotype of “advanced HF” where the biological capacity for reverse remodeling is substantially diminished. In such patients, the detrimental molecular mechanisms in cardiac remodeling, such as excessive matrix metalloproteinase activation and uncontrolled fibrosis, may have progressed to a point of irreversibility [30]. This is supported by our receiver operating characteristic (ROC) analysis, where both baseline creatinine and LVESD demonstrated strong discriminative power for predicting non-response, with AUC values comparable to that of sacubitril/valsartan treatment itself.

While pharmacotherapy with sacubitril/valsartan can powerfully modulate the response to CRT, the baseline cardiorenal and structural milieu sets the stage upon which these therapies act. Elevated serum creatinine and a larger LVESD are robust, independent markers of a myocardial substrate with a limited capacity for reverse remodeling. Future research should focus on whether targeted interventions in such high-risk patients, perhaps involving newer anti-fibrotic agents or specialized renal protective strategies, can modify this unfavorable trajectory and improve their outcomes with device-based therapy.

The functional benefits observed, specifically the superior improvement in the 6MWT and the more favorable distribution of NYHA class in the sacubitril/valsartan group, corroborate the echocardiographic and electrical findings. This triad of improved functional status, enhanced structural reverse remodeling, and superior electrical resynchronization paints a coherent picture of a therapy that acts on multiple facets of the HF syndrome. The quality of life scores (MLWHFQ and EuroQoL) improved markedly in both groups, which is a well-documented effect of CRT itself [1,31]. The fact that sacubitril/valsartan provided incremental functional benefit on top of CRT underscores its value as part of a combined pharmacological and device-based strategy.

Furthermore, clinical hard endpoints such as HF hospitalization were infrequent in our cohort, with only one event occurring in each study group. This precludes any meaningful statistical comparison or conclusion regarding the impact of sacubitril/valsartan on these endpoints, and our study was profoundly underpowered to detect differences in such rare outcomes.

### 4.1. Clinical Implications

Our results suggest that the initiation of sacubitril/valsartan should be strongly considered in all eligible patients undergoing CRT. The therapy appears to potentiate the reverse remodeling effects of biventricular pacing, leading to a higher likelihood of a positive clinical response. The potential for additional electrical and possible antiarrhythmic benefits further strengthens its role in this high-risk population. In summary, the new knowledge our study contributes is the proposition of a synergistic, drug-device combination strategy aimed at optimizing CRT response, supported by initial evidence of its association with superior multidimensional outcomes, including the novel finding of enhanced electrical resynchronization. We agree our observational design cannot prove causality, but these findings explicitly generate the hypothesis that upfront initiation of sacubitril/valsartan with CRT may be a key intervention to overcome suboptimal response, a hypothesis that now requires and is justified for testing in a dedicated randomized controlled trial.

### 4.2. Study Limitations

Several important limitations of our study must be acknowledged. First, the observational, non-randomized design introduces potential for selection bias and unmeasured confounding, despite our statistical adjustments for known prognostic factors. The treatment decision was made by the treating physicians, and while the baseline difference in ACEI/ARB use is inherent to studying sacubitril/valsartan, residual confounding cannot be ruled out.

Second, the single-center setting and modest sample size limit the generalizability of our findings. The important limitation of this study is its sample size, which, while sufficient to detect large effect sizes in key parameters of reverse remodeling, limits the power for subgroup analyses and the detection of smaller, albeit potentially clinically relevant, differences. The findings should therefore be interpreted as hypothesis-generating and require validation in larger, multi-center randomized controlled trials. Although the study was sufficiently powered to detect the large effect sizes observed in our primary reverse remodeling endpoints, it lacked the statistical power for robust subgroup analyses or to detect smaller, clinically relevant differences. A post hoc calculation indicates a power of approximately 68% (α = 0.05) for the primary response comparison, which is below the conventional 80% threshold. This underscores the need for validation in larger trials.

Third, the 12-month follow-up period was adequate to assess reverse remodeling but is too short to evaluate long-term clinical outcomes such as cardiovascular mortality or sustained reductions in heart failure hospitalization. Notably, hard clinical endpoints were rare in our cohort, with only one hospitalization event per group, rendering our study profoundly underpowered for such comparisons.

Finally, the proposed mechanistic explanations for the observed benefits, particularly those involving electrophysiological substrate modification, remain speculative and are derived from associations. Confirmation requires dedicated basic science and invasive electrophysiological studies.

We explicitly state that our findings should be interpreted as generating a promising hypothesis regarding improved reverse remodeling and clinical response at one year. They do not constitute evidence of long-term mortality benefit, which must be established in future, large-scale, multicenter randomized controlled trials with extended follow-up.

## 5. Conclusions

In patients with HF undergoing cardiac resynchronization therapy, the addition of sacubitril/valsartan to guideline-directed medical therapy is independently associated with a significantly higher rate of CRT response. This is manifested through superior improvements in functional capacity, greater left ventricular reverse remodeling, and enhanced electrical resynchronization. Our findings posit that sacubitril/valsartan works synergistically with CRT by more effectively targeting the maladaptive molecular mechanisms in cardiac remodeling that perpetuate the progression of cardiomyopathy. The potential implications for the substrate of cardiac arrhythmias, possibly through indirect effects on the myocardial milieu and ion channel homeostasis, warrant further investigation in larger, randomized controlled trials. It should be interpreted as a promising hypothesis-generating signal that requires confirmation in larger, randomized trials.

## Figures and Tables

**Figure 1 diseases-14-00006-f001:**
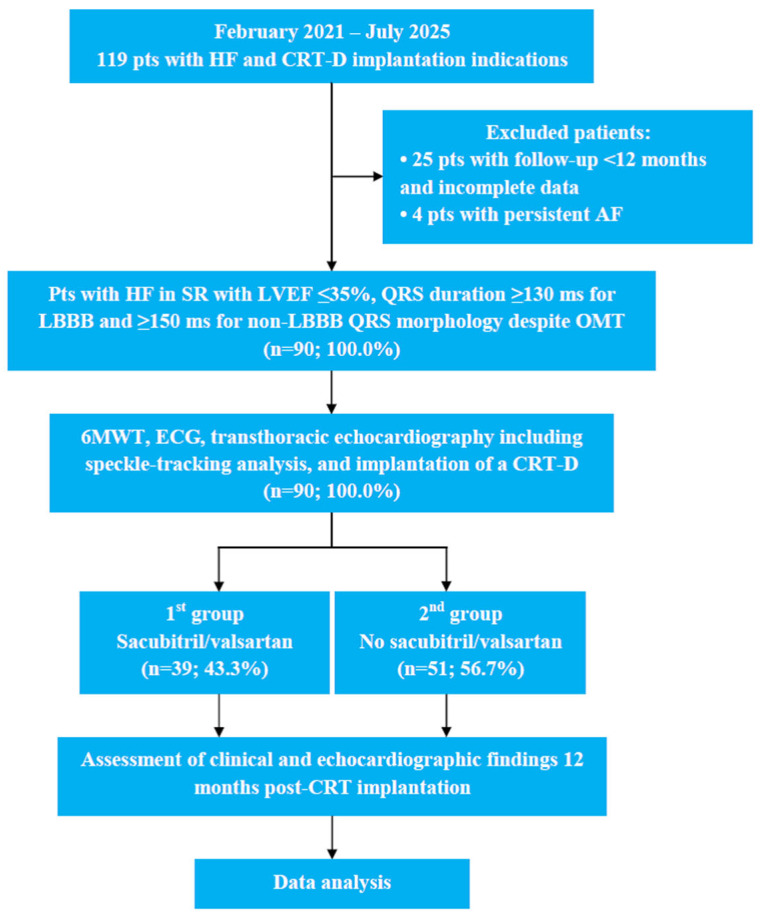
The study design and flowchart. 6MWT, 6 min walk test; AF, atrial fibrillation; BVP, biventricular pacing; CRT, cardiac resynchronization therapy; CRT-D, cardiac resynchronization therapy devices with the defibrillation function; ECG, electrocardiography; HF, heart failure; LBBB, left bundle branch block; LVEF, left ventricular ejection fraction; OMT, optimal medical therapy; SR, sinus rhythm.

**Figure 2 diseases-14-00006-f002:**
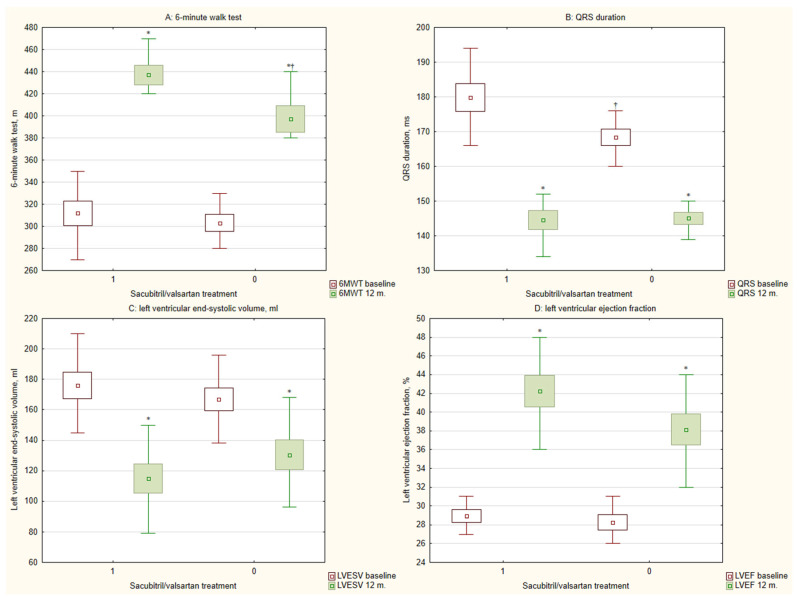
Effects of 12-month cardiac resynchronization therapy with and without sacubitril/valsartan treatment on functional (**A**), electrical conduction (**B**) and cardiac remodeling (**C**,**D**) parameters. 6MWT, 6 min walk test; LVEF, left ventricular ejection fraction; LVESV, left ventricular end-systolic volume; 1, cardiac resynchronization therapy with sacubitril/valsartan treatment; 0, cardiac resynchronization therapy without sacubitril/valsartan treatment; *, *p* < 0.001 for baseline versus 12-month comparison; †, *p* < 0.05 for within-group change from baseline.

**Figure 3 diseases-14-00006-f003:**
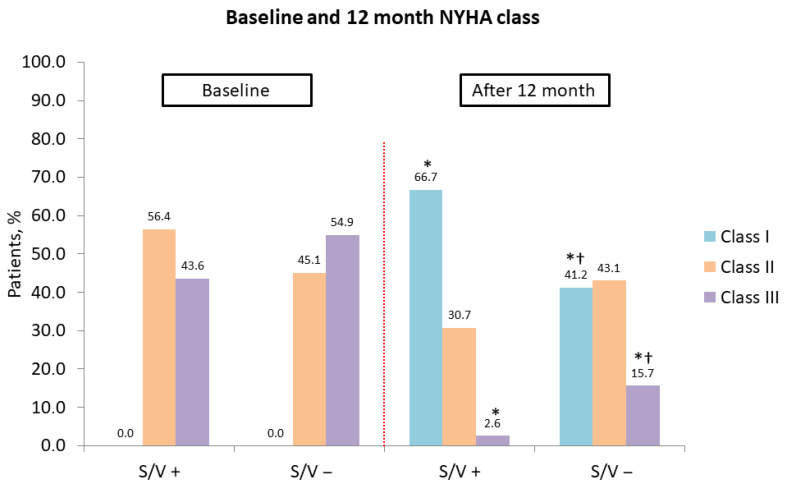
Association of cardiac resynchronization therapy and sacubitril/valsartan with improvement in New York Heart Association class at 12 months. NYHA, New York Heart Association; S/V +, cardiac resynchronization therapy with sacubitril/valsartan treatment; S/V −, cardiac resynchronization therapy without sacubitril/valsartan treatment; *, *p* < 0.001 for baseline vs. 12-month comparison; †, *p* < 0.05 for within-group change from baseline.

**Figure 4 diseases-14-00006-f004:**
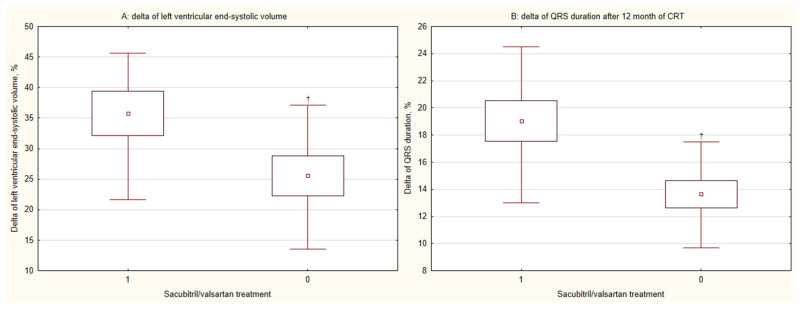
Combined effects of cardiac resynchronization therapy and sacubitril/valsartan on (A) left ventricular remodeling and (B) QRS duration at 12 months. CRT, cardiac resynchronization therapy; 1, CRT with sacubitril/valsartan treatment; 0, CRT without sacubitril/valsartan treatment; †, *p* < 0.05 for within-group change from baseline.

**Figure 5 diseases-14-00006-f005:**
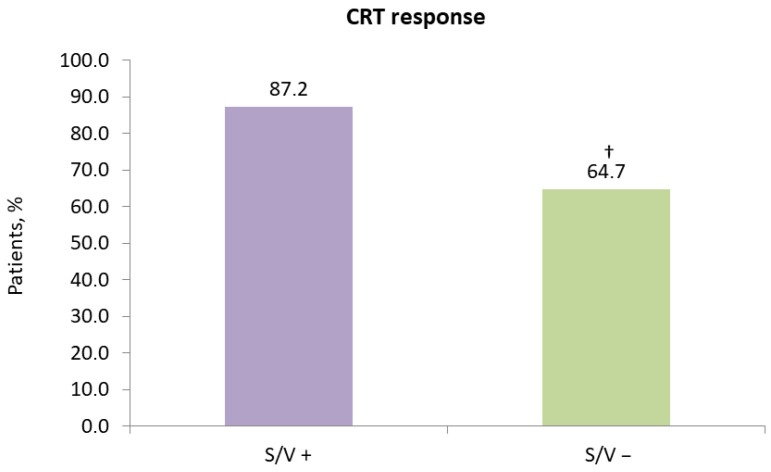
Impact of sacubitril/valsartan on 12-month response to cardiac resynchronization therapy. CRT, cardiac resynchronization therapy; S/V +, cardiac resynchronization therapy with sacubitril/valsartan treatment; S/V −, cardiac resynchronization therapy without sacubitril/valsartan treatment; †, *p* < 0.05 for within-group change from baseline.

**Figure 6 diseases-14-00006-f006:**
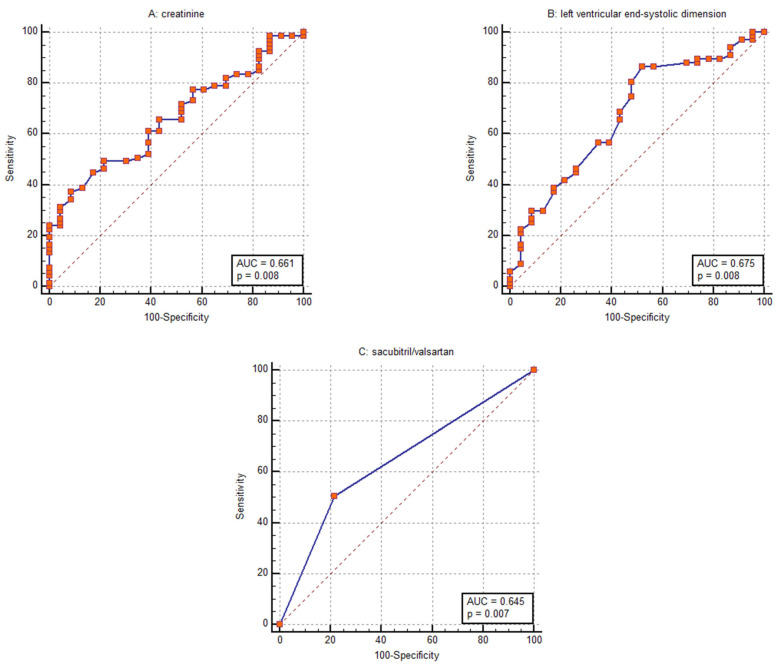
Receiver operating characteristic curves for discriminating response to cardiac resynchronization therapy using (**A**) baseline creatinine level, (**B**) baseline left ventricular end-systolic dimension and (**C**) sacubitril/valsartan treatment.

**Figure 7 diseases-14-00006-f007:**
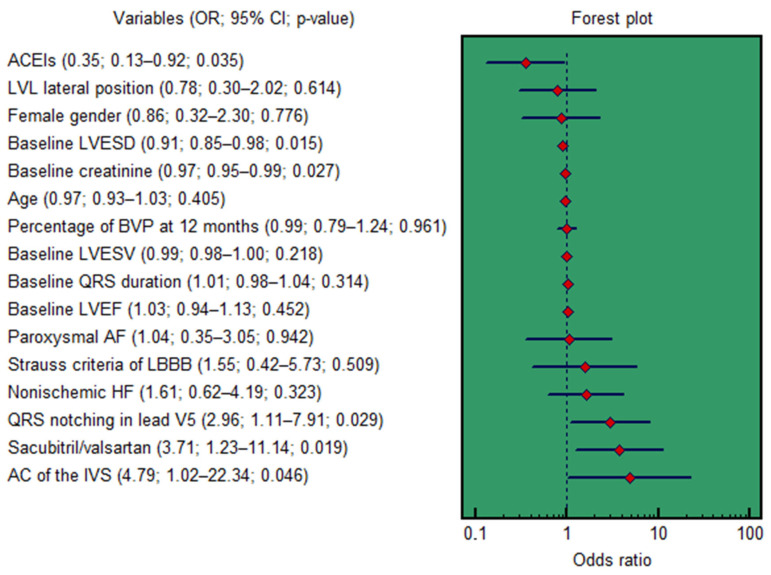
Forest plot for the univariable logistic regression results. AC, asynchronous contraction; ACEIs, angiotensin-converting enzyme inhibitors; AF, atrial fibrillation; BVP, biventricular pacing; CI, confidence interval; IVS, interventricular septum; HF, heart failure; LBBB, left bundle branch block; LVL, left ventricular lead; LVEF, left ventricular ejection fraction; LVESD, left ventricular end-systolic dimension; LVESV, left ventricular end-systolic volume; OR, odds ratio.

**Figure 8 diseases-14-00006-f008:**
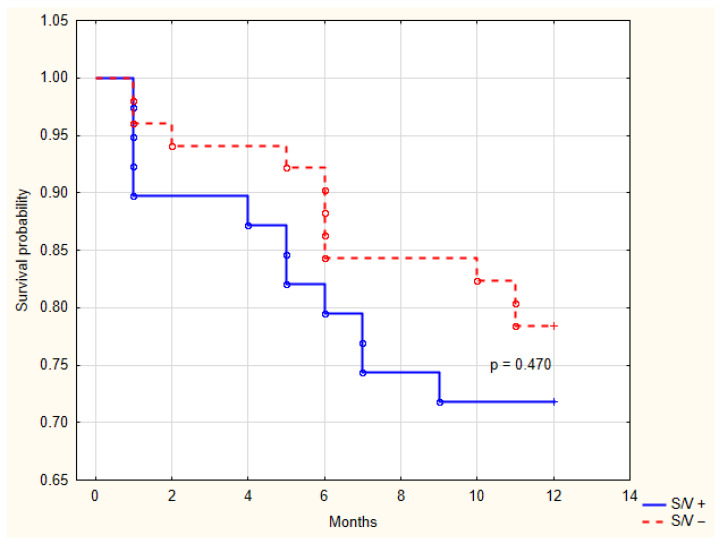
Kaplan-Meier analysis of arrhythmic events recorded by the cardiac resynchronization therapy device between groups with and without sacubitril/valsartan treatment. S/V +, cardiac resynchronization therapy with sacubitril/valsartan treatment; S/V −, cardiac resynchronization therapy without sacubitril/valsartan treatment.

**Table 1 diseases-14-00006-t001:** Baseline demographic and clinical characteristics of the overall study population and stratified groups.

Characteristics	Overall (n = 90)	1st Group S/V + (n = 39)	2nd Group S/V − (n = 51)	p_2–3_
	1	2	3	
Age, year, M ± SD	60.1 ± 9.8	60.0 ± 9.0	60.1 ± 10.4	0.596
Male gender, n (%)	57 (63.3)	25 (64.1)	32 (62.7)	0.895
Ischemic heart failure, n (%)	29 (32.2)	15 (38.5)	14 (27.4)	0.271
Non-ischemic heart failure, n (%)	51 (56.7)	20 (51.3)	31 (60.8)	0.370
Mixed etiology of heart failure, n (%)	10 (11.1)	4 (10.2)	6 (11.8)	0.822
Coronary artery stenting in anamnesis, n (%)	22 (24.4)	13 (33.3)	9 (17.6)	0.087
Body mass index, kg/m^2^, M ± SD	29.1 ± 5.0	29.8 ± 4.7	28.5 ± 5.3	0.098
Estimated GFR, mL/min/1.73 m^2^, M ± SD	71.6 ± 18.9	72.1 ± 19.8	71.2 ± 18.5	0.941
Heart failure severity (New York Heart Association functional class):				
II, n (%)	45 (50.0)	22 (56.4)	23 (45.1)	0.290
III, n (%)	45 (50.0)	17 (43.6)	28 (54.9)	0.290
Arrhythmias prior to cardiac resynchronization therapy device implantation				
Paroxysmal atrial fibrillation, n (%)	24 (26.7)	8 (20.5)	16 (31.4)	0.251
Sustained ventricular tachycardia, n (%)	5 (5.6)	2 (5.1)	3 (5.9)	0.877
Ventricular fibrillation, n (%)	2 (2.2)	2 (5.1)	0 (0.0)	0.103
Paroxysmal SVT, n (%)	2 (2.2)	0 (0.0)	2 (3.9)	0.213
Comorbidities:				
Diabetes mellitus, n (%)	21 (23.3)	12 (30.7)	9 (17.6)	0.147
Dyslipidemia, n (%)	53 (58.9)	24 (61.5)	29 (56.8)	0.656
Stroke, n (%)	5 (5.6)	2 (5.1)	3 (5.9)	0.877
Smoking, n (%)	21 (23.3)	8 (20.5)	13 (25.5)	0.582
Therapy:				
Beta-blockers, n (%)	86 (95.6)	36 (92.3)	50 (98.0)	0.193
Loop diuretics, n (%)	58 (64.4)	23 (58.9)	35 (68.6)	0.345
Mineralocorticoid receptor antagonists, n (%)	86 (95.6)	37 (94.8)	49 (96.1)	0.784
ACEIs, n (%)	33 (36.7)	0 (0.0)	33 (64.7)	<0.001
Antiplatelet agents, n (%)	49 (54.4)	22 (56.4)	27 (52.9)	0.744
Statins, n (%)	72 (80.0)	32 (82.0)	40 (78.4)	0.672
Angiotensin II receptor blockers, n (%)	18 (20.0)	0 (0.0)	18 (35.3)	<0.001
Amiodarone, n (%)	39 (43.3)	17 (43.6)	22 (43.1)	0.966
Anticoagulants, n (%)	29 (32.2)	11 (28.2)	18 (35.3)	0.478
Ivabradine, n (%)	3 (3.3)	2 (5.1)	1 (1.9)	0.409
SGLT2Is, n (%)	69 (76.7)	31 (79.5)	38 (74.5)	0.582

Values are expressed as M ± SD for continuous variables and n (%) for categorical variables. Abbreviations: ACEIs, angiotensin-converting enzyme inhibitors; GFR, glomerular filtration rate; SGLT2Is, sodium glucose co-transporter 2 inhibitors; S/V +, cardiac resynchronization therapy with sacubitril/valsartan treatment; S/V −, cardiac resynchronization therapy without sacubitril/valsartan treatment; SVT, supraventricular tachycardia. The “p_2–3_” is *p*-value between 1^st^ and 2^nd^ groups comparison.

**Table 2 diseases-14-00006-t002:** Comparative clinical and diagnostic characteristics before and after CRT-D implantation.

Characteristics	1st Group S/V + (n = 39)			2nd Group S/V − (n = 51)			p_1–4_	p_2–5_
	Baseline	12 m.	*p*	Baseline	12 m.	*p*		
	1	2	3	4	5	6		
EuroQol EQ-5D, score	57.3 ± 10.3	76.6 ± 9.6	<0.001	58.5 ± 9.5	75.1 ± 9.2	<0.001	0.363	0.277
MLWHFQ, score	59.6 ± 17.9	24.7 ± 14.8	<0.001	59.4 ± 17.7	28.3 ± 18.8	<0.001	0.829	0.579
6MWT, m	311.8 ± 70.4	436.9 ± 53.9	<0.001	303.1 ± 53.9	397.1 ± 84.0	<0.001	0.381	0.013
QRS, ms	179.8 ± 24.7	144.5 ± 16.8	<0.001	168.0 ± 19.5	146.9 ± 16.7	<0.001	0.019	0.896
∆QRS, ms	-	19.0 ± 9.5	-	-	13.6 ± 7.1	-	-	0.005
cQT_c_, ms	432.4 ± 30.4	437.1 ± 30.0	0.711	417.0 ± 38.1	437.8 ± 27.8	0.015	0.052	0.883
AA, °	−30.0 [−47.0; −5.0]	85.0 [−42.0; 135.0]	0.002	−30.0 [−47.0; 15.0]	−30.0 [−100.0; 90.0]	0.403	0.499	0.087
QRSN in V5	27 (69.2)	-	-	33 (64.7)	-	-	0.653	-
SC of LBBB	33 (84.6)	-	-	45 (88.2)	-	-	0.618	-
SBP, mmHg	118.5 ± 8.5	119.1 ± 5.8	0.010	119.6 ± 5.5	118.5 ± 5.3	0.022	0.368	0.681
DBP, mmHg	73.8 ± 4.9	74.5 ± 3.8	0.384	75.8 ± 4.2	74.8 ± 3.1	<0.001	0.059	0.944
Creatinine, µmol/L	91.9 ± 22.6	89.6 ± 24.0	<0.001	95.7 ± 21.7	98.2 ± 28.3	<0.001	0.332	0.196
LVEDD, mm	68.1 ± 6.7	62.1 ± 9.1	<0.001	67.1 ± 6.6	63.2 ± 10.1	<0.001	0.636	0.593
LVESD, mm	58.6 ± 6.7	49.5 ± 10.8	<0.001	57.8 ± 7.0	51.9 ± 12.4	<0.001	0.560	0.338
LVEF, %	28.9 ± 4.3	42.2 ± 10.5	<0.001	28.2 ± 5.6	38.1 ± 11.8	<0.001	0.711	0.068
∆LVEF, %	-	48.4 ± 40.1	-	-	37.8 ± 41.2	-	-	0.079
LVESV, ml	175.8 ± 54.6	114.9 ± 60.2	<0.001	166.8 ± 53.3	130.4 ± 70.1	<0.001	0.385	0.311
∆LVESV, %	-	35.8 ± 22.4	-	-	25.5 ± 23.2	-	-	0.025
AC of the IVS	12 (30.7)	-	-	11 (21.5)	-	-	0.324	-
Long strain, %	−8.4 ± 2.4	−12.1 ± 2.8	<0.001	−9.0 ± 2.2	−12.6 ± 3.0	<0.001	0.234	0.397
Left ventricular lead position:								
Lateral	19 (48.8)	-	-	22 (43.1)	-	-	0.600	-
Posterolateral	13 (33.3)	-	-	21 (41.2)	-	-	0.449	-
Anterolateral	7 (17.9)	-	-	8 (15.7)	-	-	0.776	-
BP rate, %	-	97.9 ± 2.7	-	-	98.1 ± 1.6	-	-	0.533

Values are mean ± SD and Me [Q1; Q3] for continuous variables and n (%) for categorical variables. Abbreviations: 6MWT, 6 min walk test; AA, alpha angle; AC of the IVS, asynchronous contraction of the interventricular septum; CRT-D, cardiac resynchronization therapy device with the defibrillator; cQTc, corrected QT interval to account for left bundle branch block or biventricular pacing; DBP, diastolic blood pressure; EuroQol EQ-5D, European Quality of Life Questionnaire; LVEDD, left ventricular end-diastolic dimension; LVEF, left ventricle ejection fraction; LVESD, left ventricular end-systolic dimension; LVESV, left ventricular end-systolic volume; MLWHFQ, Minnesota living with heart failure questionnaire; QRSN in V5, QRS notching in V5 chest lead; SBP, systolic blood pressure; SC of LBBB, Strauss criteria of left bundle branch block; S/V +, cardiac resynchronization therapy with sacubitril/valsartan treatment; S/V −, cardiac resynchronization therapy without sacubitril/valsartan treatment. The “p_1–4_” is *p*-value between baseline characteristics of 1^st^ and 2^nd^ groups comparison. The “p_2–5_” is *p*-value between 12 m. characteristics of 1^st^ and 2^nd^ groups comparison.

## Data Availability

The datasets generated and/or analyzed during this study are available from the corresponding author upon reasonable request.

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
