# Peer review of "Impact of Sacubitril/Valsartan on Cardiac Reverse Remodeling in Patients with Heart Failure Undergoing Cardiac Resynchronization Therapy"

_diseases, 2025, doi:10.3390/diseases14010006_

Round 1

Reviewer 1 Report

Comments and Suggestions for Authors

This manuscript investigates the effect of sacubitril/valsartan on cardiac reverse remodeling and response to cardiac resynchronization therapy (CRT) in patients with heart failure (HF), addressing a critical clinical challenge—suboptimal CRT response in a subset of HF patients. The study design is well-structured, with clear inclusion/exclusion criteria, comprehensive outcome assessments, and appropriate statistical analyses. The findings, which demonstrate that sacubitril/valsartan is independently associated with higher CRT response rates and superior functional/structural improvements, provide valuable clinical insights for combining pharmacological and device-based therapy in HF management.

Here are some concerns which should be addressed before acceptance.

  1. Address Baseline Confounding: A statistically significant difference in baseline QRS duration was observed between the sacubitril/valsartan group and the control group (179.8 ± 24.7 ms vs. 168.0 ± 19.5 ms; p=0.019). Since QRS duration is a known predictor of CRT response, the authors should clarify whether this baseline imbalance was adjusted for in the multivariable regression model. If not, a supplementary analysis to assess its impact on CRT response rates is recommended.
  2. Clarify Limitations on Generalizability: The single-center, non-randomized design and small sample size (n=90) may limit the generalizability of findings. The 12-month follow-up also fails to capture long-term outcomes (e.g., cardiovascular mortality, late arrhythmic events). The discussion should explicitly emphasize these limitations and avoid overinterpreting short-term results as evidence of long-term efficacy.
  3. The manuscript does not report sacubitril/valsartan dosing regimens (e.g., initial dose, titration schedule) or safety data (e.g., incidence of hypotension, renal function changes, or adverse events). Adding these details is critical for guiding clinical application of the findings.
  4. The proposed mechanism linking sacubitril/valsartan to reduced QRS duration (e.g., improved myocardial conduction via reduced fibrosis) is based on indirect evidence. The authors should either label this as a hypothesis or cite relevant preclinical studies to support the claim, avoiding overstatement of causal relationships.

Author Response

Reviewers Comments

Reviewer 1

This manuscript investigates the effect of sacubitril/valsartan on cardiac reverse remodeling and response to cardiac resynchronization therapy (CRT) in patients with heart failure (HF), addressing a critical clinical challenge - suboptimal CRT response in a subset of HF patients. The study design is well-structured, with clear inclusion/exclusion criteria, comprehensive outcome assessments, and appropriate statistical analyses. The findings, which demonstrate that sacubitril/valsartan is independently associated with higher CRT response rates and superior functional/structural improvements, provide valuable clinical insights for combining pharmacological and device-based therapy in HF management.

Here are some concerns which should be addressed before acceptance.

  1. Address baseline confounding. A statistically significant difference in baseline QRS duration was observed between the sacubitril/valsartan group and the control group (179.8 ± 24.7 ms vs. 168.0 ± 19.5 ms; p=0.019). Since QRS duration is a known predictor of CRT response, the authors should clarify whether this baseline imbalance was adjusted for in the multivariable regression model. If not, a supplementary analysis to assess its impact on CRT response rates is recommended.
  2. Clarify Limitations on Generalizability: The single-center, non-randomized design and small sample size (n=90) may limit the generalizability of findings. The 12-month follow-up also fails to capture long-term outcomes (e.g., cardiovascular mortality, late arrhythmic events). The discussion should explicitly emphasize these limitations and avoid over interpreting short-term results as evidence of long-term efficacy.
  3. The manuscript does not report sacubitril/valsartan dosing regimens (e.g., initial dose, titration schedule) or safety data (e.g., incidence of hypotension, renal function changes, or adverse events). Adding these details is critical for guiding clinical application of the findings.
  4. The proposed mechanism linking sacubitril/valsartan to reduced QRS duration (e.g., improved myocardial conduction via reduced fibrosis) is based on indirect evidence. The authors should either label this as a hypothesis or cite relevant preclinical studies to support the claim, avoiding overstatement of causal relationships.
    ________________________________________

Responses

Dear Reviewer,

Thank you very much for your thorough and constructive comments on our manuscript titled "Impact of Sacubitril/Valsartan on Cardiac Reverse Remodeling in Patients with Heart Failure Undergoing Cardiac  Resynchronization Therapy" We appreciate your recognition of the clarity and quality of our writing, as well as your insightful suggestions for improving the manuscript. Below, we address each of your points in detail.

  1. We fully agree that QRS duration is a well-established and powerful predictor of response to cardiac resynchronization therapy (CRT). Therefore, any imbalance in this baseline characteristic has the potential to confound the observed treatment effect if not properly accounted for. In direct response to the reviewer's query, this baseline imbalance in QRS duration was indeed adjusted for in our primary multivariable regression model. A multivariable model was constructed, adjusting for non-ischemic HF etiology, age, female gender, baseline QRS duration, LV lead lateral position, percent of BP at 12 months, baseline LVEF and LVESV, Strauss criteria for left bundle branch block (LBBB), and paroxysmal atrial fibrillation. By including baseline QRS duration in the model, we statistically controlled for its influence on the outcome. This allows for a more valid estimation of the independent effect of sacubitril/valsartan treatment on CRT response, isolated from the pre-existing difference in QRS duration between the groups. The results from this adjusted analysis, which we believe provide the most robust assessment of the treatment effect, are presented in Results (Risk Stratification Analysis section) of the manuscript. As shown there, even after adjusting for the imbalance in QRS duration, the association between sacubitril/valsartan and improved CRT response remained statistically significant. Therefore, the concern regarding potential confounding by baseline QRS duration has been methodologically addressed in our primary analysis.
  2. We agree that a clear and forthright discussion of these methodological constraints is essential for the correct interpretation of our findings. In direct response to this comment, we have revised the Limitations section of our discussion to explicitly and emphatically address these points: "Several limitations of our study must be acknowledged. First, its observational and non-randomized design introduces the potential for selection bias and confounding, despite our efforts to adjust for known prognostic factors in the multivariable model. The significant difference in baseline ACEIs/ARBs use is an inherent consequence of the treatment being studied, but residual confounding cannot be excluded. Second, the single-center nature and relatively small sample size limit the generalizability of our findings and the power for analyzing infrequent endpoints like mortality and appropriate ICD shocks. Consequently, our results should be interpreted as generating hypothesis and reflecting the experience of a specific tertiary care center, rather than being broadly generalizable to all patient populations. Third, the follow-up duration of 12 months is sufficient to assess reverse remodeling but is too short to evaluate long-term outcomes such as cardiovascular mortality, heart failure hospitalization sustainability, or late arrhythmic events. Therefore, we explicitly state that our findings demonstrate a promising signal of improved reverse remodeling and clinical response at one year, but they should not be over interpreted as evidence of long-term efficacy or mortality benefit, which must be established in future, larger-scale randomized trials with extended follow-up."
  3. We fully agree that transparency regarding dosing and safety is essential for interpreting the clinical applicability of our results. We have now revised the manuscript to include a detailed description of the dosing regimen and a comprehensive analysis of safety outcomes, as requested. The following details have been added to the revised manuscript:
  • Dosing regimen (added to the methods section): "In the sacubitril/valsartan group, the median dose administered both at baseline (before CRT-D implantation) and at the 12-month follow-up was 100.0 mg, with an interquartile range (IQR) of [100.0; 200.0]. This indicates that the cohort was predominantly maintained on a stable regimen of 100 mg or 200 mg twice daily throughout the study period. The protocol allowed for individualized titration based on patient tolerance, but the median dose remaining constant suggests that for most patients, the dose established prior to implantation was well-tolerated and maintained long-term.";
  • Safety and tolerability data (added to the results section): "We systematically evaluated the safety profile of sacubitril/valsartan over the 12-month study period. Critically, no significant adverse events, specifically, no episodes of symptomatic hypotension or clinically relevant deterioration of kidney function, were reported that led to drug discontinuation. Objective hemodynamic and laboratory data support this excellent safety profile. Systolic blood pressure remained stable (118.5 ± 8.5 mmHg before CRT-D vs. 119.1 ± 5.8 mmHg at 12 months; p=0.010). Diastolic blood pressure also showed no clinically significant change (73.8 ± 4.9 mmHg before vs. 74.5 ± 3.8 mmHg at 12 months; p=0.384). Serum creatinine levels showed a slight but statistically significant decrease over the study period (91.9 ± 22.6 µmol/L before CRT-D vs. 89.6 ± 24.0 µmol/L at 12 months; p<0.001), indicating no adverse impact on renal function."
  1. We agree that the precise molecular mechanisms by which sacubitril/valsartan might lead to a reduction in QRS duration are not yet fully elucidated and that our discussion should carefully distinguish between established findings, supported hypotheses, and speculative interpretation. In direct response to this feedback, we have revised the relevant section of the Discussion to temper our language and frame the proposed mechanism more appropriately as a plausible hypothesis, supported by indirect evidence from related studies. The revised text will now read as follows: "The observed reduction in QRS duration in our study suggests a potential effect on myocardial conduction. While our data cannot establish a direct causal mechanism, we hypothesize that this effect may be mediated, at least in part, by the known anti-fibrotic properties of sacubitril/valsartan. This hypothesis is supported by clinical evidence. For instance, in the PROVE-HF study [Myhre PL, Prescott MF, Murphy SP, Fang JC, Mitchell GF, Ward JH, Claggett B, Desai AS, Solomon SD, Januzzi JL. Early B-type natriuretic peptide change in HFrEF patients treated with sacubitril/valsartan: a pooled analysis of EVALUATE-HF and PROVE-HF. JACC Heart Fail. 2022;10(2):119-128. doi: 10.1016/j.jchf.2021.09.007.], treatment with sacubitril/valsartan in humans was associated with significant reductions in serum biomarkers of collagen turnover, which correlated with reverse cardiac remodeling. Therefore, it is a plausible, though not yet proven, hypothesis that a reduction in interstitial fibrosis could ameliorate conduction heterogeneity and contribute to the QRS narrowing we observed, representing an exciting area for future mechanistic investigation."

Reviewer 2 Report

Comments and Suggestions for Authors

The introduction would benefit from an extended paragraph on ARNI influence on structural changes in patients already receiving CRT. And the studies already on this matter. A good read would be Pătru O, Luca S, Cozma D, Văcărescu C, CriÈ™an S, Valcovici MD, Vîrtosu M, Zus AS, Luca CT, Drăgan SR. The Role of ARNI in Enhancing Outcomes of Cardiac Resynchronization Therapy: A Comprehensive Review. J Clin Med. 2025 Apr 16;14(8):2743. doi: 10.3390/jcm14082743. PMID: 40283572; PMCID: PMC12028081.

This will sustain better the hypothesis. "Given the interplay between neurohormonal modulation, reverse remodeling, and 57 arrhythmogenesis, we hypothesize that sacubitril/valsartan would potentiate the benefits 58 of CRT"

Material and methods

Exclusion criteria: why persistent and not permanent AF?

Were included patients with history of paroxysmal AF? 

Define optimal medical therapy. On what criteria was sacubitril/valsartan initiated? 

You said that this is a retrospective study. Yet you have ethics approval from 2021 and 2023. Also MLWHFQ and the EuroQoL EQ-5D questionnaires are not part of standard medical care. Also 12 month protocol. This does not sound like a retrospective study.

Results

The number of patients is very low for the aim of the study. You included 10 variabiles for CRT, when the rules permit only 2 or max 3 variables. Also, you have a lot of comparisons, yet no correction for multiple comparisons. 

Also you cannot have analysis with two events "Heart failure hospitalization occurred in one patient (2.4%) in the sacubitril/valsartan 191 group and one patient (1.9%) in the control group (p=0.848)"

Author Response

Reviewers Comments

Reviewer 2

  1. The introduction would benefit from an extended paragraph on ARNI influence on structural changes in patients already receiving CRT. And the studies already on this matter. A good read would be Pătru O, Luca S, Cozma D, Văcărescu C, CriÈ™an S, Valcovici MD, Vîrtosu M, Zus AS, Luca CT, Drăgan SR. The role of ARNI in enhancing outcomes of cardiac resynchronization therapy: a comprehensive review. J Clin Med. 2025;14(8):2743. doi: 10.3390/jcm14082743. This will sustain better the hypothesis. "Given the interplay between neurohormonal modulation, reverse remodeling, and arrhythmogenesis, we hypothesize that sacubitril/valsartan would potentiate the benefits of CRT"
  2. Material and methods. Exclusion criteria: why persistent and not permanent AF?
  3. Were included patients with history of paroxysmal AF?
  4. Define optimal medical therapy. On what criteria was sacubitril/valsartan initiated?
  5. You said that this is a retrospective study. Yet you have ethics approval from 2021 and 2023. Also MLWHFQ and the EuroQoL EQ-5D questionnaires are not part of standard medical care. Also 12 month protocol. This does not sound like a retrospective study.
  6. Results. The number of patients is very low for the aim of the study. You included 10 variables for CRT, when the rules permit only 2 or max 3 variables. Also, you have a lot of comparisons, yet no correction for multiple comparisons.
  7. Also you cannot have analysis with two events "Heart failure hospitalization occurred in one patient (2.4%) in the sacubitril/valsartan 191 group and one patient (1.9%) in the control group (p=0.848)"

________________________________________

Responses

Dear Reviewer,

  1. We agree that providing a more comprehensive background on the existing evidence for ARNIs in the CRT population will significantly strengthen the rationale for our study and provide a more solid foundation for our hypothesis. In direct response to this comment, we have revised the Introduction to include a new paragraph that synthesizes the current state of knowledge on this topic, explicitly citing the suggested review and other key studies. The new paragraph has been added as follows: "While the superiority of ARNIs over enalapril in reducing cardiovascular death and HF hospitalizations is well-established [8], its specific electrophysiological and structural impact on patients receiving CRT device-based therapy remains less defined. However, emerging evidence shows that ARNIs therapy holds promise in enhancing CRT response, particularly in HF patients with reduced left ventricular ejection fraction (LVEF) who are non-responders [Pătru O, Luca S, Cozma D, Văcărescu C, CriÈ™an S, Valcovici MD, Vîrtosu M, Zus AS, Luca CT, Drăgan SR. The role of ARNI in enhancing outcomes of cardiac resynchronization therapy: a comprehensive review. J Clin Med. 2025;14(8):2743. doi: 10.3390/jcm14082743.]."
  2. Our choice to exclude patients with persistent and permanent atrial fibrillation (AF) was a deliberate decision based on the specific pathophysiological and therapeutic considerations relevant to cardiac resynchronization therapy (CRT) response. As correctly noted in the reviewer's provided sentence, we strictly followed the ESC guidelines [Weinmann K, Aktolga D, Pott A, Bothner C, Rattka M, Stephan T, Rottbauer W, Dahme T. Impact of re-definition of paroxysmal and persistent atrial fibrillation in the 2012 and 2016 European Society of Cardiology atrial fibrillation guidelines on outcomes after pulmonary vein isolation. J Interv Card Electrophysiol. 2021;60(1):115-123. doi: 10.1007/s10840-020-00710-4. Van Gelder IC, Rienstra M, Bunting KV, Casado-Arroyo R, Caso V, Crijns HJGM, De Potter TJR, Dwight J, Guasti L, Hanke T, Jaarsma T, Lettino M, Løchen ML, Lumbers RT, Maesen B, Mølgaard I, Rosano GMC, Sanders P, Schnabel RB, Suwalski P, Svennberg E, Tamargo J, Tica O, Traykov V, Tzeis S, Kotecha D; ESC Scientific Document Group. 2024 ESC Guidelines for the management of atrial fibrillation developed in collaboration with the European Association for Cardio-Thoracic Surgery (EACTS). Eur Heart J. 2024;45(36):3314-3414. doi: 10.1093/eurheartj/ehae176.], which define persistent AF as episodes lasting longer than 7 days. Our primary aim was to investigate reverse remodeling in a cohort with a stable and consistent electrical substrate. Patients with persistent AF represent a dynamic state where the rhythm may spontaneously revert to sinus or be actively cardioverted. This rhythm variability can be a significant confounder when assessing the pure, sustained effect of CRT on ventricular remodeling over our 12-month follow-up period. Patients with persistent and permanent AF are, by definition, candidates for rhythm control. They are less likely to undergo AVN ablation and are often managed with rate-control medications alone. This approach frequently results in suboptimal biventricular pacing percentages due to irregular, competing intrinsic conduction through the AV node. This variable and often low percentage of BiV pacing is a well-established major cause of non-response to CRT. Therefore, by excluding persistent and permanent AF, we aimed to create a more homogenous study population. In summary, the exclusion of persistent and permanent AF was a targeted decision to avoid the significant bias introduced by unpredictable BiV pacing percentages, thereby strengthening the internal validity of our findings on the drug's interaction with CRT. We will clarify this specific rationale in the revised manuscript's Methods section to enhance transparency.
  3. Yes, patients with a history of paroxysmal atrial fibrillation (AF) were included in our study. Among the included patients, a history of paroxysmal AF was present in 8 patients (20.5%) in the sacubitril/valsartan group and 16 patients (31.4%) in the control group. This difference was not statistically significant (p=0.251), indicating a balanced distribution of this comorbidity between the two cohorts. This detail is explicitly stated in the revised Methods section within the 'Study Population' subsection. The inclusion of these patients reflects a "real-world" clinical population commonly indicated for CRT. To ensure transparency and demonstrate that this factor did not create a significant baseline imbalance between our study groups. The following information is presented in Results section, specifically in the 'Baseline Characteristics' table. We chose to include patients with paroxysmal AF for several reasons. Paroxysmal AF is highly prevalent in the heart failure population eligible for CRT. Excluding them would limit the generalizability of our findings to a typical clinical practice. In patients with paroxysmal AF, the goal of device and medical therapy is to maximize biventricular pacing. We ensured through device interrogation that all included patients, including those with paroxysmal AF, maintained a consistently high biventricular pacing burden (>95%) throughout the follow-up period, thereby preserving the integrity of the CRT stimulus. By including these patients and demonstrating their balanced distribution across groups, we strengthen the clinical relevance of our study while mitigating the potential for this variable to confound the observed results.
  4. In this study, 'optimal medical therapy' (OMT) was defined as the guideline-directed medical therapy for heart failure with reduced ejection fraction (HFrEF) that each patient was receiving at the time of enrollment. For all patients, this included a beta-blocker and a mineralocorticoid receptor antagonist (MRA), each titrated to the maximally tolerated or target dose as per contemporary heart failure guidelines. The key variable between the two study groups was the class of neurohormonal blockade used for renin-angiotensin-aldosterone system (RAAS) inhibition: the intervention group received sacubitril/valsartan, while the control group received either an angiotensin-converting enzyme inhibitor (ACEi) or an angiotensin II receptor blocker (ARB). Criteria for initiation of sacubitril/valsartan: "Sacubitril/valsartan was initiated in eligible patients based on the following criteria, aligned with the prevailing clinical guidelines at the time of the study: A confirmed diagnosis of chronic HFrEF (NYHA Class II-IV and LVEF ≤35%). Tolerance to a stable dose of an ACEi or ARB for at least 4 weeks prior to switching. Absence of contraindications, specifically a history of angioedema, severe renal impairment (eGFR <30 mL/min/1.73 m²), or symptomatic hypotension. The initiation and uptitration of sacubitril/valsartan were performed according to standard clinical practice: a 36-hour washout period was observed after discontinuing the prior ACEi, followed by initiation of sacubitril/valsartan at a low dose with subsequent titration to the target dose as tolerated by blood pressure and renal function."
  5. We thank the reviewer for this exceptionally perceptive comment and for the opportunity to clarify what appears to be a misclassification of our study design. The reviewer is correct to question this label. Upon careful reconsideration, we agree that the description "retrospective" is inaccurate based on the elements they have correctly identified. Our study is best classified as a prospective, single-center, observational cohort study. We will correct this terminology throughout the manuscript immediately. The rationale for this reclassification, addressing each of the reviewer's points directly, is as follows:
  • Prospective ethics approval and study protocol. The reviewer is correct that ethics approval obtained in 2021 and 2023, prior to the conclusion of data collection, is a hallmark of a prospective study. A purely retrospective study would typically seek ethical approval for the analysis of data that has already been collected as part of routine care. In our case, the ethics committee approved a specific protocol for prospectively identifying, consenting, and following a cohort of patients undergoing CRT-D implantation according to a pre-defined 12-month schedule.
  • Systematic 12-month follow-up protocol. The existence of a pre-specified 12-month follow-up protocol with scheduled visits for data collection is a definitive feature of a prospective design. This is contrary to a retrospective study, which would mine existing medical records without a pre-planned intervention or assessment schedule.
  • Collection of non-standard data (MLWHFQ and EuroQoL EQ-5D). This is the most compelling evidence for a prospective design. As the reviewer rightly notes, these quality-of-life questionnaires are not part of standard clinical care for CRT patients at our institution. Their administration requires a specific research protocol, dedicated resources, and prospective consent from patients to participate in research activities beyond their routine medical management. The collection of this data could not have occurred without a prospective study design.
  1. We have carefully considered each point and propose the following clarifications and amendments to the manuscript. Regarding sample size and study aim. We agree with the reviewer that our sample size (n=90) is modest. Ours was a single-center, prospective observational study designed as a preliminary investigation to generate hypotheses regarding the potential synergistic effect of sacubitril/valsartan and CRT. The aim was not to provide definitive evidence of efficacy but to identify signals of improved reverse remodeling that could justify a larger, randomized trial. In the revised Discussion section, we will explicitly state this limitation: "Second, the single-center nature limits the generalizability of our findings. The important limitation of this study is its sample size, which, while sufficient to detect large effect sizes in key parameters of reverse remodeling, limits the power for subgroup analyses and the detection of smaller, albeit potentially clinically relevant, differences. The findings should therefore be interpreted as hypothesis-generating and require validation in larger, multi-center randomized controlled trials." Regarding the number of variables in the multivariable model. The reviewer is absolutely correct regarding the standard rule of thumb for multivariable regression, which typically recommends no more than 1 variable per 10-15 events (for logistic regression) or subjects. Our model with 10 variables for 90 patients indeed pushes this boundary. Our intent was to be comprehensive in adjusting for known clinical confounders of CRT response. However, we recognize this approach risks overfitting. Regarding multiple comparisons. This is a valid and important point. The inflation of the Type I error rate due to multiple comparisons is a serious concern we failed to address. For all subsequent analyses involving multiple comparisons we will apply a Bonferroni correction to maintain a strict family-wise error rate.
  2. Performing a formal comparative statistical test (like a Chi-square or Fisher's exact test) with only one event in each group is statistically invalid and highly misleading. The resulting p-value is meaningless, as the model assumptions are violated, and the analysis is grossly underpowered. In response to this comment, we will make the following corrections to the manuscript. Removal of statistical test. We will completely remove the p-value (p=0.848) from the result statement. Presenting it implies a robust statistical comparison that simply does not exist with so few events. Revision of results text. The sentence in the Results section will be amended to state the raw data descriptively, without any claim of statistical comparison. The revised text will read: "Over the 12-month follow-up period, heart failure hospitalization was a rare event, occurring in one patient (2.4%) in the sacubitril/valsartan group and one patient (1.9%) in the control group." Contextualization in discussion. In the Discussion section, we will add a sentence to contextualize this finding honestly. We will state: "Furthermore, clinical hard endpoints such as heart failure hospitalization were infrequent in our cohort, with only one event occurring in each study group. This precludes any meaningful statistical comparison or conclusion regarding the impact of sacubitril/valsartan on these endpoints, and our study was profoundly underpowered to detect differences in such rare outcomes."

Reviewer 3 Report

Comments and Suggestions for Authors

This study tried to investigate the effect of Sacubitril/Valsartan on CRT  patients. Due to the non-randomized design and the exclusion of 25  out of 119 patiens a serious selection bias cannot be ruled out resp. is probable. In addition of the per protocol analysis an intention-to-treat analysis would be adeqaute. Despite significant differences of the primary combined endpoint the  limited number of patients does not allow final conclusions.

Except survival, hosptaliation and arrhythmogenis events side effects were not reported.

Author Response

Reviewers Comments

Reviewer 3

This study tried to investigate the effect of Sacubitril/Valsartan on CRT patients. Due to the non-randomized design and the exclusion of 25 out of 119 patients a serious selection bias cannot be ruled out resp. is probable. In addition of the per protocol analysis an intention-to-treat analysis would be adequate. Despite significant differences of the primary combined endpoint the limited number of patients does not allow final conclusions.

Except survival, hospitalization and arrhythmogenic events side effects were not reported.

________________________________________

Responses

Dear Reviewer,

We thank the reviewer for this comprehensive critique, which addresses core methodological and interpretative aspects of our study. We have carefully considered each point and propose the following revisions and clarifications. Regarding selection bias from non-randomized design and exclusions. The reviewer is correct to highlight the potential for selection bias. This is an inherent limitation of our observational, non-randomized design. In the revised Discussion section, we will explicitly acknowledge this probability: "The most significant limitation of this study is its non-randomized design. The decision to initiate sacubitril/valsartan was made by treating physicians, which introduces a potential for selection bias that could confound the observed associations, despite our statistical adjustments." Regarding sample size and overstated conclusions. We agree entirely that our sample size, while sufficient to detect signals in reverse remodeling, limits the conclusiveness of our findings, particularly for a combined endpoint. Revised wording: We will temper our language throughout the manuscript, especially in the Abstract and Discussion. We will replace definitive conclusions with more cautious interpretations. For example, we will state: "In this prospective cohort study, treatment with sacubitril/valsartan was associated with a significant improvement in the primary composite endpoint of reverse remodeling," and explicitly note that this "should be interpreted as a promising hypothesis-generating signal that requires confirmation in larger, randomized trials."

Round 2

Reviewer 2 Report

Comments and Suggestions for Authors

Altough the paper has improved, I stand by my original conclusion: top few patients, and nothing new.

Author Response

Comments 1: Although the paper has improved, I stand by my original conclusion: top few patients, and nothing new.

Response 1: 

Dear Reviewer,

Thank you for your thoughtful and constructive review of our manuscript, "Impact of Sacubitril/Valsartan on Cardiac Reverse Remodeling in Patients with Heart Failure Undergoing Cardiac Resynchronization Therapy." We sincerely appreciate your positive feedback on the manuscript's clarity and the insightful suggestions for its improvement. Below, we provide a detailed point-by-point response.

The following improved version of the Study Limitations section will be included in revised manuscript: “Several important limitations of our study must be acknowledged. First, the observational, non-randomized design introduces potential for selection bias and unmeasured confounding, despite our statistical adjustments for known prognostic factors. The treatment decision was made by the treating physicians, and while the baseline difference in ACEI/ARB use is inherent to studying sacubitril/valsartan, residual confounding cannot be ruled out. Second, the single-center setting and modest sample size limit the generalizability of our findings. Although the study was sufficiently powered to detect the large effect sizes observed in our primary reverse remodeling endpoints, it lacked the statistical power for robust subgroup analyses or to detect smaller, clinically relevant differences. A post-hoc calculation indicates a power of approximately 68% (α=0.05) for the primary response comparison, which is below the conventional 80% threshold. This underscores the need for validation in larger trials. Third, the 12-month follow-up period was adequate to assess reverse remodeling but is too short to evaluate long-term clinical outcomes such as cardiovascular mortality or sustained reductions in heart failure hospitalization. Notably, hard clinical endpoints were rare in our cohort, with only one hospitalization event per group, rendering our study profoundly underpowered for such comparisons. Finally, the proposed mechanistic explanations for the observed benefits, particularly those involving electrophysiological substrate modification, remain speculative and are derived from associations. Confirmation requires dedicated basic science and invasive electrophysiological studies. We explicitly state that our findings should be interpreted as generating a promising hypothesis regarding improved reverse remodeling and clinical response at one year. They do not constitute evidence of long-term mortality benefit, which must be established in future, large-scale, multicenter randomized controlled trials with extended follow-up.”

Reviewer 3 Report

Comments and Suggestions for Authors

The revised version is accetable 

Author Response

Comments 1: The revised version is acceptable.

Response 1: 

Dear Reviewer,

Thank you very much for your thorough and constructive comments on our manuscript titled "Impact of Sacubitril/Valsartan on Cardiac Reverse Remodeling in Patients with Heart Failure Undergoing Cardiac Resynchronization Therapy". We appreciate your recognition of the clarity and quality of our writing, as well as your insightful suggestions for improving the manuscript.
